# Stochastic disturbance regimes alter patterns of ecosystem variability and recovery

**Jennifer M. Fraterrigo**[1,2]*, **Aaron B. Langille**[3], **James A. Rusak**[4,5]

**1** Department of Natural Resources and Environmental Sciences, University of Illinois, Urbana, Illinois, United States of America, **2** Program in Ecology, Evolution and Conservation Biology, University of Illinois, Urbana, Illinois, United States of America, **3** Department of Mathematics and Computer Science, Laurentian University, Sudbury, Ontario, Canada, **4** Dorset Environmental Science Centre, Ontario Ministry of the Environment, Conservation and Parks, Dorset, Ontario, Canada, **5** Department of Biology, Queen's University Dorset, Dorset, Ontario, Canada

* jmf@illinois.edu

**Data Availability Statement:** All relevant data are within the manuscript and its Supporting Information files.

## Abstract

Altered ecosystem variability is an important ecological response to disturbance yet understanding of how various attributes of disturbance regimes affect ecosystem variability is limited. To improve the framework for understanding the disturbance regime attributes that affect ecosystem variability, we examine how the introduction of stochasticity to disturbance parameters (frequency, severity and extent) alters simulated recovery when compared to deterministic outcomes from a spatially explicit simulation model. We also examine the agreement between results from empirical studies and deterministic and stochastic configurations of the model. We find that stochasticity in disturbance frequency and spatial extent leads to the greatest increase in the variance of simulated dynamics, although stochastic severity also contributes to departures from the deterministic case. The incorporation of stochasticity in disturbance attributes improves agreement between empirical and simulated responses, with 71% of empirical responses correctly classified by stochastic configurations of the model as compared to 47% using the purely deterministic model. By comparison, only 2% of empirical responses were correctly classified by the deterministic model and misclassified by stochastic configurations of the model. These results indicate that stochasticity in the attributes of a disturbance regime alters the patterns and classification of ecosystem variability, suggesting altered recovery dynamics. Incorporating stochastic disturbance processes into models may thus be critical for anticipating the ecological resilience of ecosystems.

## Introduction

Understanding and anticipating ecosystem responses to disturbance is a fundamental goal in ecology and conservation that has become even more pressing in the face of global environmental changes that alter disturbance regimes. Although most theoretical and empirical studies focus on mean responses, there is increasing recognition that patterns of variability also contain valuable information about disturbance effects [1, 2]. For example, ecologists have shown that changes in variability can indicate differences in the predictability of responses [3,

**Funding:** This work was partially supported by the National Science Foundation (US) (DEB-1637522). Funding support was provided by the Cooperative Freshwater Ecology Unit at Laurentian University. The funders had no role in study design, data collection and analysis, decision to publish, or preparation of the manuscript.

**Competing interests:** The authors have declared that no competing interests exist.

4], recovery time [5, 6], and the internal behavior of an ecosystem [7]. Shifts in the variability of an ecological response can also signal changes in ecosystem recovery dynamics and impending ecosystem state transitions [8–10], thereby providing insights into the ecological resilience of an ecosystem, i.e., the magnitude of disturbance that a system can absorb before it changes stable states [11]. Monitoring such ecological resilience indicators may enable actions to avert these outcomes [12, 13]. The ability to predict how natural and anthropogenic disturbances affect variability is therefore of great interest, particularly from a policy perspective, where patterns of variability are increasingly being used to guide decision-making [14–16] and to promote the ecological resilience of ecosystems in a management context [17].

Disturbances can alter patterns of ecosystem variability for several reasons. Within and among ecosystems, spatial and temporal variation in the physical environment and biota modulate disturbance effects and recovery [18, 19]. For example, species' responses to and recovery from disturbance events depend on their life history, dispersal capacity, and competitive ability [20–23]. Additionally, disturbance regimes are typically stochastic and differ with respect to frequency, severity, and extent, as well as other properties [24]. Variation along any of these axes can subsequently result in different recovery dynamics [25, 26].

Given the inherent complexity of changes in even one of these attributes, it is common to turn to qualitative conceptual models for insight, especially when attempting to generalize across systems of differing properties. Such models can help refine and prioritize questions for further empirical work, as well as inform additional modeling efforts. These conceptual models can also be implemented quantitatively to provide a more explicit understanding of how alternative formulations of key processes influence responses to altered drivers, thus generating testable predictions for future work e.g., [27–29]. Of particular importance to the work on the relationship between disturbance and ecosystem variability is the model developed by Turner et al. [30], which sought to identify the spatial and temporal attributes of a disturbance regime that could lead to qualitatively different landscape dynamics. Turner et al. [30] characterized landscape dynamics by computing the constancy of seral stages (i.e., temporal variability in the amount of cover) under varying combinations of disturbance spatial extent and frequency at a range of scales. The model provides a solid and tractable baseline for discussion of the effects of disturbance regime properties on ecosystem dynamics. However, because the model is deterministic, it does not address how stochasticity in disturbance regime attributes affect ecosystem variability and resultant dynamics. Disturbances are only partially deterministic, and a constrained model may obscure relationships between the properties of disturbance regimes and ecosystem variability. Understanding the effects of stochastic disturbance regimes on ecosystem variability is even more important under a changing environment where altered disturbance regimes will be the norm rather than the exception [24].

Previously, we used the fully deterministic framework of Turner et al. [30] to predict how disturbance might alter the variability of terrestrial and aquatic ecosystems [1]. We compared model predictions with the results of published studies that documented changes in ecosystem variability following disturbance. Although we found good overall agreement between predictions and observations, many outcomes were not as expected. We highlighted the need to incorporate more realistic estimates of species recovery as well as the difficulties associated with properly characterizing chronic disturbance events in the framework as possible improvements, but did not elaborate on the role that stochasticity might play in improving the agreement between the model and empirical studies.

To address this gap, we extended the deterministic model of Turner et al. [30] by modifying the attributes of disturbance regimes to include stochasticity in three principle disturbance parameters: frequency, severity and spatial extent, and investigated which stochastic disturbance parameters might be most influential in generating changes in ecosystem variability and

recovery dynamics. We also explored whether stochastic parameterizations improve the agreement between simulated dynamics based on the deterministic and extended stochastic model and results from published empirical studies. Our work demonstrates that accounting for stochasticity in the attributes of a disturbance regime significantly affects the patterns and classification of ecosystem variability, suggesting altered ecosystem dynamics.

## Methods

### Base model

We duplicated the deterministic model of Turner et al. based on their description of the parameters and processes [30]. The model environment consisted of a set of $100 \times 100$ gridded cells. Eight vegetation classes representing successional stages were included in the model. Each cell within the grid had a variable that tracked the successional stage and the modelled environment began with all cells at the most mature stage (successional stage 8). The model used a discrete time representation; at each time step a disturbance could be introduced into the environment and any previously disturbed grid cell could recover. Disturbances could occur in all successional stages, were of a fixed size with four equal sides ($s$) and were imposed at a fixed temporal frequency ($f$). In this base model all disturbed regions recovered deterministically by passing through the eight successional stages (one per time step) until the mature stage was reached. Eight is therefore also the value we use for ecosystem recovery time (see "S and T parameter space" below). Although Turner et al. [30] created the model to mimic successional dynamics in a plant assemblage, they also encouraged its extension to other ecosystems.

Disturbance spatial extent and frequency have been shown to strongly affect recovery dynamics [7, 20, 30]. In the model, the spatial extent ($s^2$) determines the number of cells affected when a disturbance event occurs. The location of the center of the disturbance extent is determined by selecting coordinates randomly from a uniform distribution. Frequency ($f$) determines the number of disturbance events per time step during a simulation, and is inversely related to disturbance return interval ($1/f$), which is the number of time steps between successive disturbance events. Disturbances are evenly spaced, every $1/f$ time steps throughout the duration of a simulation, and in the case of multiple events per time step (when $f > 1$), each is executed during that time step (see "Stochastic Frequency" below for further explanation and examples). In the deterministic base model, all disturbances within a simulation use the same spatial-extent and frequency parameters (i.e., the values are fixed). The spatial boundaries of the environment are considered toroidal such that any disturbances that extend beyond the horizontal or vertical limits "wrap" around to the opposite side, eliminating boundary effects. Each model time step proceeded as follows: (1) increase any previously disturbed grid locations by one successional stage (i.e., recover); (2) if a new disturbance event is scheduled, determine center of disturbance randomly; (3) set all newly disturbed grid locations to lowest successional stage. This latter step represents the modelled severity (i.e., impact on the ecosystem) and, in the deterministic base model, a disturbance always resets a grid cell back to the lowest successional stage.

We extended this deterministic model through the addition of stochasticity into all three key disturbance properties: frequency, severity, and spatial extent, as described below. To maintain generality, all parameter values are drawn from a pseudorandom uniform distribution. While some observable phenomena may be more accurately replicated using a more specific distribution (normal, exponential, Weibull, etc.), a uniform random selection provided a reasonable starting point for extending the base model and analyzing its sensitivity.

## Stochastic frequency

It is often difficult to predict specifically when a disturbance event will occur because initiating events are typically random (e.g., lightning strike), although over time the average number of events can often be predicted with reasonable accuracy. To ensure comparability with the deterministic model, we maintained the same average number of events over the duration of a simulation, as determined by the frequency parameter *f*. Before the simulation was run, a time-line was established and disturbance events were scheduled on randomly selected time steps. Consider a simple example with a deterministic frequency of one event every 6 time steps for simulation with 100 time steps. In comparison with the deterministic base model scenario, 16 distinct, evenly spaced events will occur. In our extended stochastic model, the 16 scheduled events will occur but are unlikely to be evenly spaced given the random selection process. Instead they may be clustered or potentially occur within the same time step, as events are drawn with replacement.

## Stochastic severity

Like frequency, the severity of a disturbance event was also modelled stochastically. The deterministic model assumes that all disturbance events "reset" a location on the landscape to the lowest successional stage (successional stage 1). While this is reasonable under some conditions, it is not always realistic. For example, fire severity spans a gradient from stand-replacing crown fires to a lightly burned or scorched event [31], and a single disturbance can result in the manifestation of all of these burn severities. Here we implemented stochastic disturbance severity by differentially adjusting the successional stage of a cell. We randomly selected a disturbance severity in each time step and a disturbance was imposed such that the disturbed locations were set back to a successional stage lower than the current stage but not necessarily successional stage 1 (and never lower than successional stage 1). The same level of severity was applied across the entire disturbance extent if more than a single event occurred at the same time. As a result, cells differed in their susceptibility to disturbance. For example, a cell at successional stage 2 would not be "reset" if disturbance severity was $\geq 2$. This is analogous to a recovering cell being "skipped" by disturbance. Further, because individual disturbance events could have low or high severity, simulations executed with stochastic severity had lower mean severity than simulations with deterministic severity.

## Stochastic spatial extent

Similarly, the spatial extent of a disturbance event is often stochastic. A deterministic disturbance extent is likely for only a limited number of cases (harvesting, controlled burn, etc). We simulated stochastic spatial extent by randomly selecting the width and length (within the 2D environment) of the disturbance from a uniform distribution. The width and length of each event are calculated independently and can range from 1 cell to 2 times the maximum dimension specified in the deterministic model. For example, a parameterization of the deterministic model for disturbance events with spatial extent 20 × 20 cells produces multiple, identically sized disturbances of 400 cells. In our extended model, each disturbance event would have dimensions between 1 cell and 40 cells for each side, producing disturbances with a spatial extent of 420 cells on average. Thus, each event has a unique rectangular shape and size, but the average size of the events over the duration of the simulation are comparable to the corresponding deterministic parameterization.

## Fully stochastic

Disturbance regimes are likely to occur on a continuum ranging from largely deterministic disturbances (e.g., spruce budworm infestations, prescribed burning and other forest management practices) to those that are random in frequency, severity and spatial extent (e.g., extreme storm events, floods). Further, because disturbances events can co-occur and interact, we also confronted the model with stochasticity in all parameters by applying the above protocols to all variables simultaneously.

## Simulation setup and computations

**Model validation.**    To validate the model and permit visual inspection of recovery dynamics under different model configurations, each simulation was initially run for 100 time steps with deterministic or stochastic parameters (S1 Table). Under the deterministic configuration, disturbance frequency was set to 0.125, resulting in 8 time steps between events, and disturbance extent was $35 \times 35$ units, while severity was parameterized such that each disturbance reset the successional stage of the disturbed area to 1. We re-parameterized the model for stochastic execution with frequency, spatial extent, and severity of the disturbances simulated separately and in combination as described above. Results were expressed as the mean and variance of the proportion ($Vp$) of the landscape occupied by the mature successional stage (successional stage 8) over the duration of the simulation. This provides a measure of stability, defined as constancy in structural attributes [32, 33] and forms the basis of our comparisons of ecosystem variability and dynamics in deterministic and stochastic scenarios. The results of model validation are reported in S1 Appendix.

**S and T parameter space.**    To provide general insights about how disturbance regime attributes affect patterns of ecosystem variability and compare across ecosystems and disturbance regimes which can vary greatly in spatial and temporal dynamics, the spatial extent and return interval of disturbance events were expressed relative to the size of the environment and its temporal recovery interval respectively. Specifically, the spatial parameter (S) is disturbance size ($s^2$) / ecosystem size (where ecosystem size is constant at 10,000 (100 x 100 units) and the temporal parameter (T) is disturbance return interval ($1/f$) / ecosystem recovery time, where recovery time is the number of time steps required for a disturbed cell to achieve the "mature" stage (8 times steps). Using these two key parameters, we defined the state space characterizing the response of ecosystem variability and recovery dynamics to various disturbance regimes.

To compare the effects of stochasticity in this S-T space, we configured and ran the deterministic and stochastic models to generate a wide range of S and T values. Using a factorial approach, we quantified the mean and variance resulting from all combinations of S and T for values of S ranging from 1–100 in steps of 1, and values of T ranging from 0.01–1000 in steps of $10^i$, where $3 \geq i \geq -2$ and varies by -0.1, for a total of 50 steps. The small incremental values of S and T enabled us to quantify the mean and variance across the state space at a high resolution (S2 Table). For each unique combination of S and T, we performed a simulation for 100,000 time steps to compute a statistically robust mean and variance. Generally, the model stabilized in < 100 time steps, so the influence of transient dynamics on the calculated metrics (i.e., mean and variance) was minimal.

The results of these simulations were used to identify regions within the S and T parameter space that suggested qualitatively different dynamics based on quantitative patterns in the mean and variance of the proportion of the landscape occupied by the mature successional stage [*sensu* 30]. These regions represent the range of observed or theoretically predicted dynamics, including steady state dynamics [34, 35], stable dynamics with repeating recovery

sequences characterized by low or high response fluctuations [36, 37], and unstable dynamics [38]. The definition of each region is as follows: region A, where the landscape is relatively undisturbed, the mature stage is dominant ($> 50\%$ of the landscape) and variance is low ($< 5$), indicating equilibrium dynamics; region B, where disturbance is more frequent, the mature stage is dominant and variance is low (5–10), indicating stable dynamics with low variance; region D, where disturbance is still more frequent, the mature stage is $< 50\%$ of the landscape and variance is low (5–10), indicating stable dynamics with low variance; and regions C and E, where coverage of the mature stage fluctuates substantially producing high (10–20) or very high ($> 20$) variance, respectively, indicating stable dynamics with high variance; and region F, where disturbances are both frequent and large, producing low variance ($< 5$) and unstable dynamics, as the mature stage is no longer dominant over the duration of the simulation. Turner et al. [30] suggested that disturbance regimes that place ecosystems in region F might result in state changes.

To evaluate changes in the patterns of variability from stochastic disturbance regimes, we quantified differences in the values of $Vp$ produced by deterministic and stochastic simulations to highlight the parameter space where deviations were largest. We also evaluated how the classification of recovery dynamics changed by comparing the number of pixels falling within the ranges of S and T that defined each qualitatively unique region A-F among the state-space diagrams. For all simulations, the environment was defined as a $100 \times 100$ grid and the recovery interval was characterized by 8 successional stages. To explore the sensitivity of the model to the number of successional stages and differences in mean severity, we repeated the simulations above using a model with 4 and 16 successional stages.

## Comparison of published empirical results and simulated results

We tested whether incorporating stochasticity improved the agreement between empirical results reported in the literature (S2 Table) and results simulated by the deterministic model and our extended model. Studies were located through electronic searches of the Web of Science (Clarivate Analytics) citation index using the keywords: disturbance, heterogeneity, stability, variance and variability, and by examining the references in these studies. Although these search methods may have missed studies, those included here represent a wide range of disturbances, ecosystems and scales, and thus allow for broad inference and testing of our model. We restricted the studies to those that focused on structural (cover, abundance) or compositional responses to disturbance and recorded how the mean and variance changed in response to disturbance to align with the types of responses simulated by the model. We found that the mean did not change much between our models; thus, we focused on variance as the response of greatest interest because it signals where dynamics may shift qualitatively. In sum, we identified 45 suitable empirical cases that focused on either forest (6), grassland (9), freshwater (12), or marine (18) ecosystems (S2 Table).

To classify the response of ecosystem variability and dynamics to the disturbance regime described in each empirical study, we executed deterministic and stochastic simulations using the base and extended models. Models were parameterized using the reported values for disturbance return interval and extent, and five simulations were performed for each study. The first simulation was fully deterministic, with all disturbance parameters having fixed values. In the next three simulations, one disturbance parameter varied stochastically (frequency, severity, or extent) while all other parameters were fixed. For simulations with stochastic extent or frequency, we incorporated stochasticity as described above, by drawing values from a uniform distribution with restrictions to ensure that the average parameter value over all times steps was comparable to the value reported in the study being considered (see "Stochastic

Frequency" and "Stochastic Spatial Extent"). For simulations with stochastic severity, severity was randomly selected as described above (see "Stochastic Severity"). In the final simulation, all three disturbance parameters could vary stochastically. We ran each simulation for 100,000 time steps and calculated the mean and variance (*Vp)* of the proportion of the landscape occupied by the mature successional stage over the duration of the simulation.

The mean and variance computed from each simulation were used to characterize qualitative patterns of ecosystem dynamics for each study and evaluate data-model agreement. As in our previous work [1], we focused on qualitative rather than quantitative patterns to allow comparison across ecosystems and disturbances which can vary greatly in spatial and temporal scales, and to provide insights into conditions under which disturbance produces state shifts in ecosystem dynamics. We used the mean and *Vp* to classify the dynamics produced by a given simulation as either: equilibrium (A); stable, low variance and dominated by the mature stage (B); stable, low variance and not dominated by the mature stage (D); stable, high variance (C/E combined); or unstable, low variance (F), as described above (see "S and T parameter space"). We likewise classified the dynamics observed in each of the empirical studies based on the reported changes in the mean and variance of the response following disturbance. For example, if we found a minor change in the mean and a major change in the variance, then we classified the study as 'stable/high variance (C/E).' To evaluate data-model agreement, we compared the number of studies correctly classified by a deterministic model configuration, a stochastic model configuration, both a deterministic and stochastic model configuration, or incorrectly classified by all models using a chi-square goodness of fit test, where the expected frequency of each category was identical (i.e., 0.25) to indicate an equal chance of being in one of the four groups. Analyses were performed using R statistical software [39].

To facilitate comparison among the different studies, we also computed S and T parameters for each study via the estimation of ecosystem extent and recovery interval using the information supplied in the study. For example, in Coleman et al. [40] variability of marine algal cover increased substantially following release from grazing but algae remained the dominant cover type. The perturbation occurred over half of the experimental area at a rate of once per month, with an estimated recovery of three months. We thus coded the dynamics as "stable, high variability (C/E)" with S = 0.5 and T = 0.3; see S3 Table. When more than one response was examined in a study, we evaluated them separately, apart from responses that had identical values for S and T and showed qualitatively equivalent changes in variance.

## Results

### S and T parameter space

Under deterministic conditions, the mean coverage of the mature successional stage varied with S and T. When the interval between disturbances was long relative to recovery time (i.e., high T) and/or disturbance spatial extent was small relative to landscape extent (low S), the mature stage dominated (> 50% coverage; Fig 1). Overall, the mature stage dominated across 56% of the state space defined by S and T. Other stages dominated when disturbance spatial extent was large relative to landscape extent (higher S) and relatively frequent (lower T).

Implementing stochastic parameters had modest effects on the dominance of the mature stage (Fig 1b–1d). Similar to deterministic simulations, stochastic simulations resulted in the mature stage dominating across 57, 57, 61 and 62% of the state space for stochastic frequency, stochastic extent, stochastic severity, and fully stochastic simulations, respectively. Comparing the mean values derived from deterministic and stochastic simulations, we found that the coverage of the mature stage increased when disturbance extent was large and of intermediate frequency (Fig 2). Smaller and more frequent disturbances also increased the coverage of the

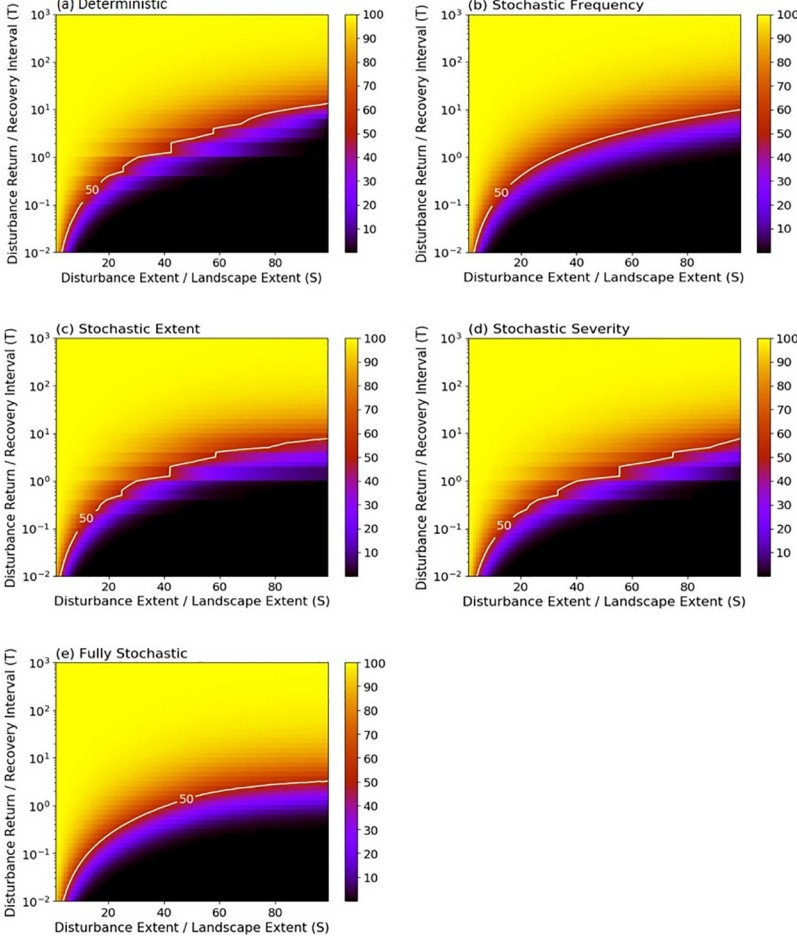

**Fig 1. Mean coverage of the mature successional stage produced by deterministic and stochastic simulations.**
Scenarios include a) deterministic, b) stochasticity in disturbance frequency, c) stochasticity in disturbance spatial
extent, d) stochasticity in disturbance severity, and e) stochasticity in all disturbance parameters: frequency, spatial
extent and severity. Colors indicate the values of mean coverage of the mature successional stage, which were
calculated under varying disturbance regime characteristics summarized by S (expressed as % of the landscape) and T.
In the region above the solid white line, the mature successional stage dominates (> 50% of the landscape); in the
region below the solid line, the other successional stages dominate.

mature successional stage, but only under stochastic severity and fully stochastic scenarios (Fig
2).

In contrast, stochasticity in disturbance parameters had large effects on variance patterns.
Comparing the values of $Vp$ derived from deterministic and stochastic simulations, we found
that the variance of simulated dynamics increased most when disturbance regimes had inter-
mediate frequencies and large spatial extents (Fig 3). The smallest changes occurred when dis-
turbance regimes had a relatively small extent, and either low or high frequency. This is likely
because when disturbances are small or occur very infrequently, there is less potential for
interactions between disturbances, or when they are very frequent, the system is almost always
at the lowest successional stage. Stochastic severity decreased the mean severity of the distur-
bance regime and generally caused more modest changes in variance compared to other sto-
chastic attributes (Fig 3).

Comparing a cross-section from the simulations (see Fig 3) at a fixed spatial extent (35 x
35) across the full range of *f* values represented by disturbance return interval (1/*f*), we found

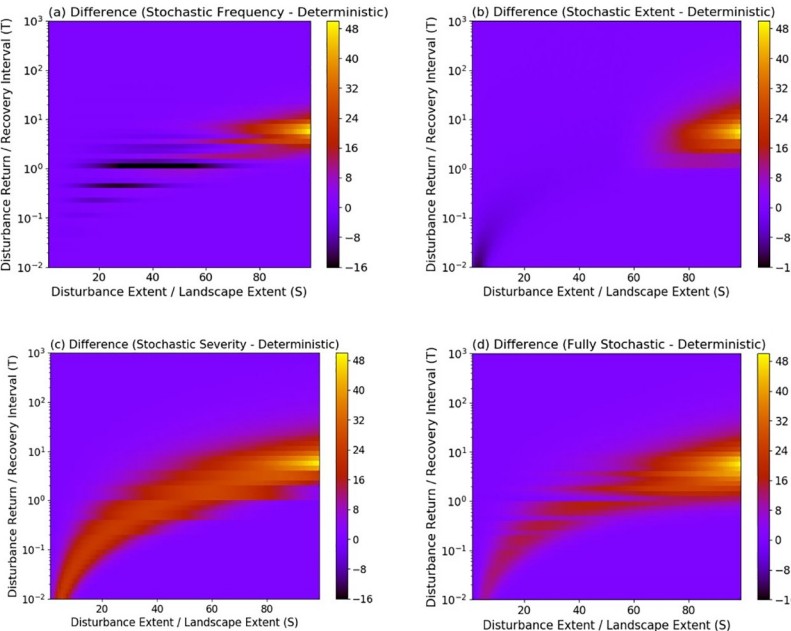

**Fig 2. Quantitative changes in the mean coverage of the mature successional stage from stochastic model configurations compared to deterministic model configurations.** Changes are characterized as the difference in mean coverage of the mature successional stage between stochastic and deterministic simulations, under varying disturbance regime characteristics. Colors represent the magnitude of differences in mean coverage. The spatio-temporal attributes of the disturbance regimes producing the differences are summarized by S (expressed as % of the landscape) and T.

that variance in the occupancy of the mature successional stage diverged the most for values of disturbance return interval between 0.1 and ~7 (Fig 4). Between these values, stochastic disturbance frequency and spatial extent resulted in higher variance than stochastic disturbance severity, and all these configurations produced higher variance than deterministic parameterizations. Increasing $f$, which equates to decreasing disturbance return interval, caused a decline in variance because the landscape was increasingly dominated by early successional stages. Decreasing $f$, which equates to increasing disturbance return interval, caused a rapid decline in variance for all parameter configurations because the landscape was increasingly dominated by the late successional stage. Under these conditions, however, stochastic disturbance extent produced higher variance because of the potential for disturbances to vary in size and to overlap with previous disturbance events, thus causing a greater delay in recovery to the mature successional stage. As we increase or decrease disturbance return time to their extremes, variance decreases as the mature successional stage is either dominant due to lack of disturbance or perpetually in a state of recovery due to multiple overlapping disturbances respectively. However, all models with stochastic elements, show a maximized variance around when disturbance return interval ⊡ 1; i.e., when a single disturbance occurs every time step. At this value, the mature successional stage vacillates between the greatest range of recovery and disturbance conditions, thus producing the highest variance.

One other notable result was the behavior of the fully deterministic version, which showed a sharp decrease in variance as disturbance return interval approached 8 and then a spike in variance as it exceeded 8 (Fig 4). This behavior emerged as an artifact of the deterministic model. Since the number of successional stages and thus ecosystem recovery was set to 8, when 1/$f$ ⊡ 8, variance went to zero because a new disturbance occurred just as the landscape

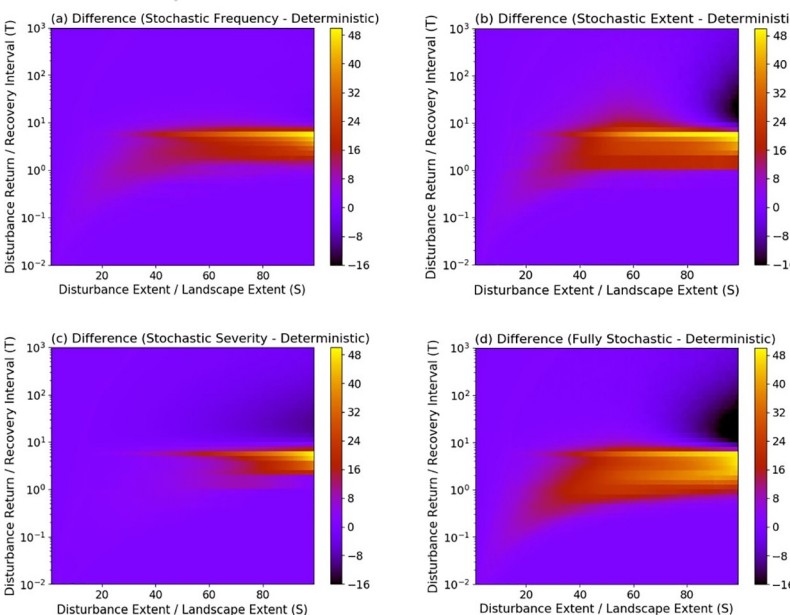

**Fig 3. Quantitative changes in the variance of the mature successional stage from stochastic model configurations compared to deterministic model configurations.** Changes are characterized as the difference in $Vp$ between stochastic and deterministic simulations, where $Vp$ is the variance in occupancy of the mature successional stage under varying disturbance regime characteristics. Colors represent the magnitude of differences in $Vp$. The spatio-temporal attributes of the disturbance regimes producing the differences are summarized by S (expressed as % of the landscape) and T.

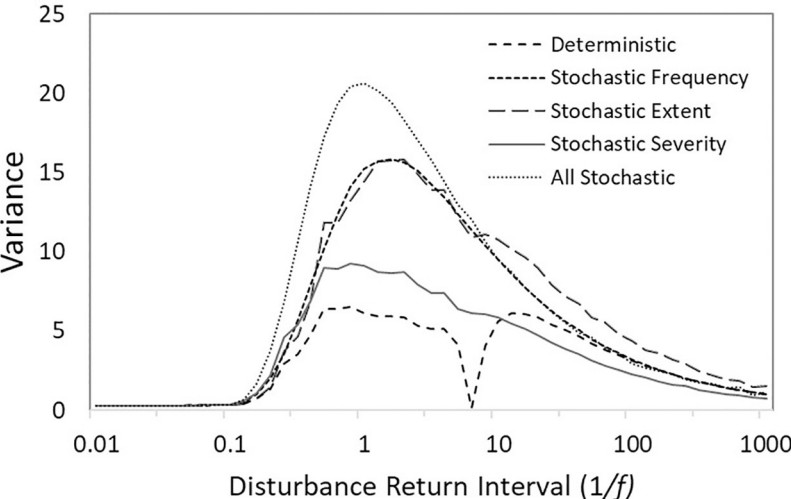

**Fig 4. Effects of changing disturbance frequency on variance in occupancy of the mature successional stage for different sets of model parametrizations.** For all simulations, disturbance frequency ($f$) is plotted on the x-axis. In the deterministic parameterization, all parameters are fixed, with disturbance extent = 35 × 35 cells, and disturbance severity is maximal (reset to 1 when disturbed). In the remaining parameterizations, the manipulated parameter is stochastic and all other parameters are fixed with values used in the deterministic parameterization.

neared full recovery. As a result, the proportion of landscape at successional stage 8 was nearly constant over time. This behavior occurs only in the deterministic model and only when the disturbance return interval equals ecosystem recovery time.

Stochasticity also altered the domains of the S-T parameter space where qualitatively different dynamics emerged (Fig 5). Compared to the regions derived from deterministic simulations (Fig 5a), the regions of the parameter space with stable but high variance (region C) or very high variance (region E) expanded when stochastic parameters were applied (Fig 5b–5e). The size of region C increased by 58, 62, 29, and 55% and the size of region E increased by 130, 111, 28, and 142% for stochastic spatial extent, stochastic frequency, stochastic severity, and fully stochastic simulations, respectively. Thus, the size of regions associated with high variance nearly doubled under most stochastic scenarios. By contrast, all other regions contracted: the size of the region characterized by stable, low variance (B and D combined) was 8–11% smaller, the region characterized by equilibrium (region A) was 1–28% smaller, and the region

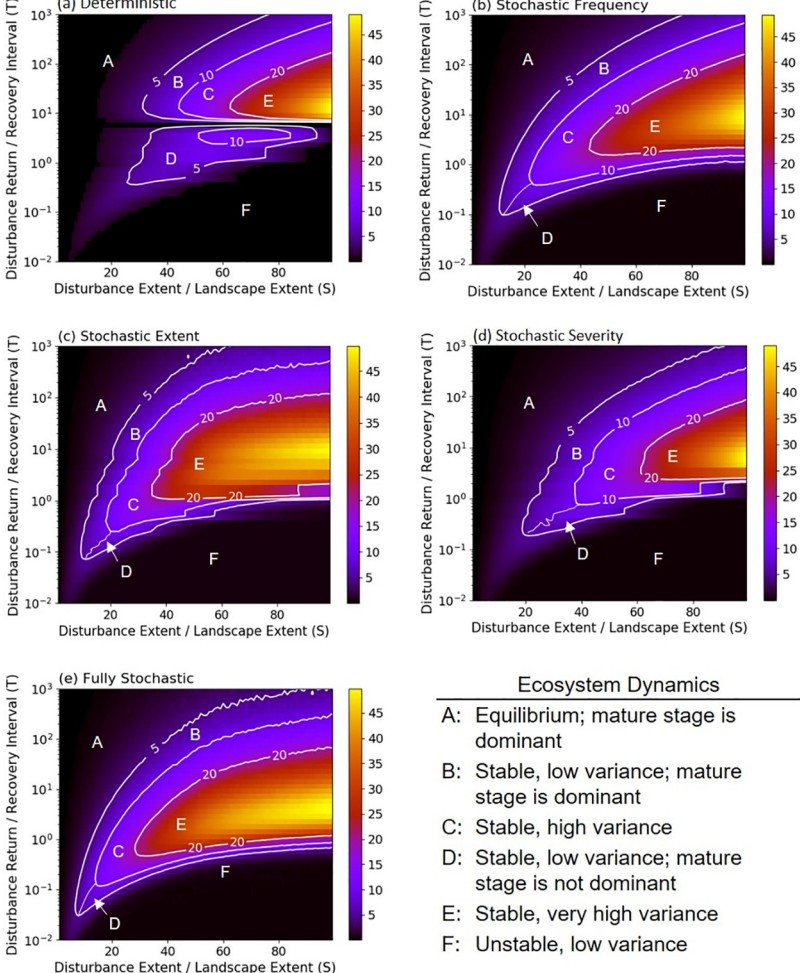

**Fig 5. Ecosystem dynamics produced by deterministic and stochastic simulations.** Dynamics are characterized by the mean and variance in occupancy of the mature successional stage under varying disturbance regime characteristics. Colors indicate the values of variance in occupancy of the mature successional stage ($Vp$) over the duration of the simulation and contours indicate the transitions in ecosystem dynamics associated with spatio-temporal changes in disturbance regimes as summarized by S (expressed as % of the landscape) and T.

characterized by unstable, low variance (region F) was 8–19% smaller. For both regions A and F, the smallest change in size occurred when we implemented stochastic severity. The simulations producing the greatest decrease in region size differed by region. For region A, implementing stochastic disturbance extent caused the greatest decrease in region size. For region F, incorporating stochasticity into all the disturbance parameters resulted in the greatest decrease in region size.

Overlaying the classified regions shown in Fig 5, we found that the classification of ecosystem dynamics changed the most for stochastic disturbance regimes at intermediate disturbance frequencies (Fig 6). The largest shifts were from states characterized by stable, low variance (regions B and D) to those characterized by stable, high (region C) or very high variance (region E). For the transition between low to high variance states, the most pronounced differences resulted from stochasticity in single parameters. For the transition between low to very high variance states, the most pronounced differences resulted from stochasticity in all disturbance parameters (Fig 6). Stochasticity in all disturbance parameters resulted in relatively larger shifts from the unstable, low variance state (region F) to stable, low (region D), high (region C) or very high variance (region E) states.

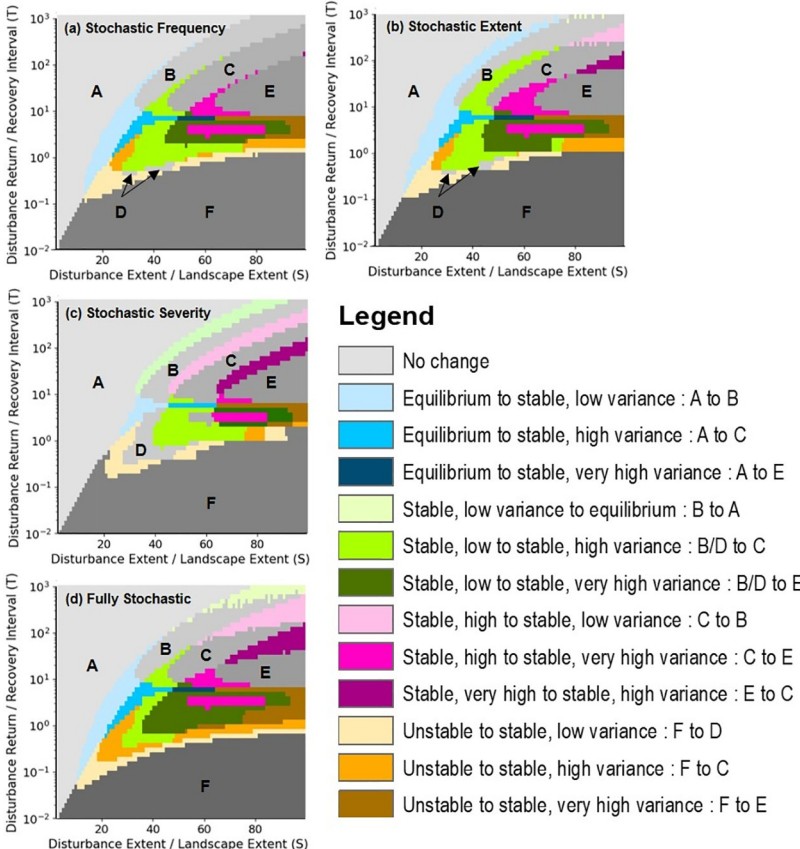

**Fig 6. Transition maps indicating where shifts in the classification of ecosystem dynamics resulted from stochastic versus deterministic model configurations.** Ecosystem dynamics were characterized with respect to the mean and variance in occupancy of the mature successional stage over the duration of the simulation and relative to the S (expressed as % of the landscape) and T parameter space. Comparisons are made between the fully deterministic version of the model and (a) stochastic frequency, (b) stochastic spatial extent, (c) stochastic severity, and (d) all stochastic disturbance parameters. Gray shades indicate no change in state, and colors represent transitions between states.

When we explored the sensitivity of the model to different numbers of successional stages, we found minimal differences in the sizes of the regions or domains of the parameter space where qualitatively different ecosystem dynamics emerged (Fig 5e and S2 Fig). The main effect of changing the number of successional stages was that mean severity changed concurrently. The simulations had mean severities of 1.5, 3.5, and 7.5 for models with 4, 8, and 16 successional stages, respectively.

## Comparison of published empirical results and simulated results

We evaluated a total of 45 published empirical responses; 35 of these were previously examined by Fraterrigo and Rusak (2008) and 10 were from new studies (S3 Table). While disturbance extent varied among the studies, over 60% of had S = 1 when normalized with respect to the size of the system. Comparing empirical results with simulated results, we found that 47% of the responses were correctly classified by both the deterministic and stochastic models (Table 1). Of the remaining 24 responses, 46% (11/24) were correctly classified by one of the stochastic configurations and misclassified by the fully deterministic model, indicating an improvement in data-model agreement and an overall revised model agreement of 71% (32/45; S3 Table). Stochastic configurations of the model correctly classified the variability and dynamics of studied ecosystems more often than expected if cases were evenly divided among the four categories in Table 1 ($\chi^2$ = 40.7, p < 0.001).

The stochastic models were successful at classifying both low and high variance dynamics (Table 1). The fully stochastic configuration accounted for most of the improvement in classification (11 cases), followed by configurations with stochastic frequency, extent, severity, or (7 cases each). Improved classification of ecosystem dynamics generally occurred for disturbance regimes that had values of S and T that placed them near the boundaries of regions in the state space. This was especially true for cases where the deterministic model predicted that a disturbance regime would cause unstable dynamics (region F) whereas the stochastic model configuration predicted stable dynamics with high variance (region C or E; S3 Table, section 1). One empirical response was correctly classified by the deterministic model and misclassified by stochastic model, and 27% of the responses were misclassified by all model configurations.

**Table 1. Summary of results from an analysis evaluating the correspondence between empirical results reported in the literature and simulated results from stochastic and deterministic models.**

| Observed response (region) | % of responses correctly classified by stochastic models and misclassified by deterministic model (observed / predicted) | % of responses correctly classified by deterministic model and misclassified by stochastic models (observed / predicted) | % of responses correctly classified by both stochastic and deterministic models (observed / predicted) | % of responses misclassified by both stochastic and deterministic models (observed / predicted) |
|---|---|---|---|---|
| All responses | 24 (11/45) | 2 (1/45) | 47 (21/45) | 27 (12/45) |
| Stable, high variance (C or E) | 22 (8/37) | 3 (1/37) | 54 (20/37) | 22 (8/37) |
| Stable, low variance (B or D) | 40 (2/5) | 0 (0/5) | 0 (0/5) | 60 (3/5) |
| Equilibrium (A) | 33 (1/3) | 0 (0/3) | 33 (1/3) | 33 (1/3) |
| Unstable, low variance (F) | - | - | - | - |

## Discussion

Disturbance-driven changes in variability are increasingly recognized as ecologically important, yet our conceptual framework for understanding shifts in variability is incomplete [1, 3]. Here, we found that regimes with stochastic disturbance frequency or extent most often led to an increase in ecosystem variance and altered ecosystem dynamics at intermediate frequencies and large spatial extents (Figs 3 and 5). Stochastic severity contributed to modest increases in variance compared to the baseline deterministic case (Fig 3). The incorporation of stochastic disturbance regimes increased the agreement between empirical and simulated results, as evidenced by an increase in the number of studies whose results were correctly classified by stochastic model configurations (Table 1). This increased agreement was partly the result of an overall increase in the size of the S and T parameter space associated with higher variability (Figs 5 and 6). Given that few disturbance regimes are fully deterministic in nature, these outcomes have important implications for understanding how disturbance will affect ecosystem behavior.

In their seminal work relating characteristics of disturbance regimes to landscape dynamics, Turner et al. [30] concluded that disturbance frequency and extent are critically important for determining system stability. In an extension of this conclusion, we found that the largest increases in the variance of the proportion of the landscape occupied by the mature successional stage over all time steps occurred when stochastic disturbance regimes were characterized by intermediate frequency and large spatial extents (Fig 3). At intermediate frequencies of disturbance, the probability that a landscape will be dominated by the mature successional stage or any single successional stage is low. Most of the system is in different phases of recovery, and this results in large fluctuations in structural attributes over time (Fig 4). This pattern is most prominent when disturbance size is large because more of the system is changing asynchronously. Stochasticity amplifies these effects by increasing the likelihood that disturbances will interact, leading to more patches on distinct successional trajectories. At high or low levels of disturbance frequency, the landscape is dominated by early or late successional stages (Fig 1), respectively, and variance in the occupancy of the mature successional stage declines (Fig 4). Under low disturbance frequency (i.e. long disturbance return interval) however, stochasticity in disturbance extent causes a modest increase in system variance through spatial disturbance interactions, which can convert systems from an equilibrium state to a stable, low or high variance state (Fig 6). Using spatially uniform (analytical) models simulating biomass density, Zelnik et al. [23] likewise found that the combination of spatial and temporal dimensions of disturbances can have large effects on ecosystem stability but spatial interactions between disturbed areas contributed to high variance under high rather than low disturbance frequency. This disparity underscores the fact that different measures of stability may respond differently to disturbance regimes.

Our finding that stochastic frequency and spatial extent increased variance in ecosystem dynamics most when compared to deterministic disturbance parameters agree with a recent meta-analysis of the predictability of community response to disturbance which showed that random disturbances generally led to decreased predictability in abundance [3]. For example, disturbances that create open habitat for new colonists have low response predictability, potentially because variation in dispersal, a largely stochastic process, causes differences in how recovering communities assemble [7]. Likewise, variation in distance to propagule sources can lead to alternate successional pathways [41–43]. Stochastic frequency and resultant clustering of disturbance events can also increase the likelihood that an area will be disturbed again before it has fully recovered, which can reduce the abundance of surviving organisms and propagules, substantially decreasing the predictability of succession [44, 45]. Models that

represent these processes are needed to distinguish among potential mechanisms driving variance and recovery patterns [23].

Model configurations with stochastic frequency and extent also altered the spatial and temporal domains of the parameter space where qualitatively different ecosystem dynamics emerged (Figs 5 and 6). In particular, the regions in which stable, high or very high variance is expected were larger when these parameters were stochastic. This is because stochastic disturbance frequency and extent increase temporal fluctuations in the abundance of individual successional stages. Despite this, ecosystem dynamics remained stable, with the mature successional stage dominating across more than half of the parameter space (Fig 1). Moreover, stochasticity did not increase the size of the region where unstable, low variance ecosystem dynamics prevail (region F). This result indicates that stochastic disturbance regimes will not always drive systems to collapse in finite time. Martín et al. [46] showed mathematically that spatial heterogeneity can reduce the possibility of catastrophic state change. While we lack the empirical evidence needed to directly test these hypotheses, recent studies investigating changes in species composition and biome boundary shifts (e.g., [24]) suggest that more frequent disturbance events will lead to higher ecosystem variance. In tropical South America, for example, increased fire frequency is expected to cause a substantial reduction in tree cover and expand the forest-savanna transitional area, under less favorable climate conditions [47]. Similarly, an increased frequency of drought events in a sub-tropical lake was coincident with increased variance in net ecosystem production in the lake [48]. Storm events are also predicted to increase their frequency and extent under a changing climate and to the extent this will affect both terrestrial and aquatic ecosystems, the modelled increase in variance suggests an impairment of our ability to predict ecosystem response [4].

In contrast, simulations with stochastic severity resulted in a smaller increase in variance compared to the deterministic model and shifts in ecosystem dynamics from states characterized by high variance to those characterized by low variance (Figs 3 and 5). As implemented, stochastic severity resulted in disturbances that did not always fully reset the community to the earliest successional stage. Consequently, some disturbed areas recovered to the mature stage more rapidly (i.e., in fewer time steps). This suggests that variation in post-disturbance survival could result in greater temporal stability. In fact, biotic legacies are increasingly viewed as important for promoting ecological resilience to disturbance [49]. However, simulations with stochastic severity also yielded lower severity on average over all time steps compared to deterministic simulations (S1 Table). Although we found that this had only modest effects on patterns of ecosystem variance and recovery dynamics (S2 Fig), additional research is needed to disentangle the effects of stochastic severity from those of lower mean severity on ecosystem variance.

We found that the simulation models incorporating stochasticity correctly classified empirical responses more often than deterministic simulations. Correct classification increased from 47% to 71% when some form of stochasticity was introduced into the purely deterministic model. Improved classification of ecosystem dynamics mainly occurred for disturbances with values of S and T that placed them near the boundaries of regions in the state space, particularly for cases where the deterministic model predicted that a disturbance regime might cause unstable dynamics, but the stochastic model configuration predicted stable dynamics with very high variability. This suggests that accounting for stochasticity better constrains the disturbance conditions that are expected to lead to ecosystem instability. Stochastic simulations may improve predictions of how disturbance regimes affect ecosystem dynamics because they account for reduced predictability of structural properties. Supporting this idea, Murphy and Romanuk (2012) found that disturbance commonly resulted in reduced predictability of structural but not diversity-related responses.

That said, our modelled vs. empirical validation is not without its caveats. Many (over 60%) of the studies used in the set of published results have a disturbance size of 1, where the entire landscape has undergone disturbance. While not ideal from a testing perspective, this is the reality of the published literature on disturbance and variability. Although it is conceivable that researchers have a bias toward studying disturbances of large spatial extent, this may also truly reflect the distribution of disturbance size found in nature. Certainly, aquatic environments are typically disturbed at the whole ecosystem scale due to their spatial homogeneity. Further, given the necessary requirements for inclusion in this comparison, the total number of applicable studies was not large, and the resultant power of our comparison is diminished. Nonetheless, an overall classification improvement of nearly 25% is noteworthy and highlights the need for an increased consideration of variability as a response in studies of ecosystem dynamics.

Ultimately, any model, no matter how complex, offers a simplified view of ecosystems and our refinements to the deterministic model of Turner et al. [30] are no exception. We offer a simple model of ecosystem dynamics based on a step-wise theory of succession that has since been replaced by more sophisticated and nuanced theories [50–52] and no way for competition, dispersal, or legacy effects to operate in the context of succession dynamics [49, 53, 54]. However simple models that can be largely substantiated by empirical results [55] and possess an ability to predict outcomes from observations not used in model design and parameterization [56] can be very powerful tools. Interaction strengths can often be weaker than expected [57, 58], competitive interactions are often not dominant in many ecosystems [59, 60], and the dynamics of species replacement, whether successional or not, certainly operate in ecosystems–both terrestrial [50] and aquatic [61]. In this context, the simplicity of our model allows us to demonstrate that basic attributes of disturbance can generate potentially diverse outcomes in ecosystem variability. Those studies that do not agree with predicted changes in system variability offer the potential for insights into additional mechanisms that may prove to be important in incorporating into an improved framework for forecasting the effects of disturbance on ecosystem dynamics. One promising direction for future studies could include alteration of the degree of stochasticity incorporated into disturbance attributes (i.e., the range over which frequency, extent, and severity can vary). For example, using a spatially explicit simulation model, Fraterrigo et al. [27] evaluated how different levels of environmental stochasticity influenced population dynamics on landscapes that varied in habitat configuration and found that higher levels of stochasticity increased variability in population size. Incorporating a greater range of stochasticity may therefore substantially shift regions in S and T space and capture a larger percentage of studies, thus increasing our understanding of the role that these individual attributes play in altering the ecological resilience of ecosystems.

Understanding the effects of stochastic disturbance regimes on ecosystem variability should provide important practical information to natural resource managers. Managers have appreciated the vital role of disturbances in shaping ecosystem dynamics for some time, relying on natural variability concepts to guide contemporary ecosystem management [14, 15]. Our results provide new insights into the disturbance conditions under which altered dynamics can be expected. Specifically, our results suggest that stochastic disturbance regimes will often yield stable, but high variance dynamics. However, because human activities and climate change are altering environmental conditions, disturbance effects are changing [24]. For example, fire and windthrow have triggered rapid changes in tree community composition in southern boreal and northern hardwood forests across central North America where climate change is shifting the prairie-forest ecotone [62]. Consequently, more sophisticated approaches can build upon these outcomes to improve quantitative predictions about how the characteristics of disturbance regimes will affect ecosystem response variability under anthropogenic environmental change.

## Supporting information

**S1 Table. Parameters and results for the deterministic and stochastic simulations used to validate the model and explore ecosystem variability and recovery dynamics.**
(DOCX)

**S2 Table. Spatial (S) and temporal (T) scales of disturbance used to create state-space diagrams illustrating potential disturbance dynamics.**
(DOCX)

**S3 Table. Empirical studies evaluated for data-model agreement.**
(DOCX)

**S1 Fig. Occupancy by successional stage for simulations with deterministic and stochastic disturbance parameters.**
(DOCX)

**S2 Fig. State-space diagrams describing potential ecosystem dynamics for fully stochastic (i.e., stochastic spatial extent, frequency, and severity) models with 4, 8, or 16 successional stages with respect to temporal and spatial disturbance regime attributes.**
(DOCX)

**S1 Appendix. Model validation results.**
(DOCX)

**S2 Appendix. References for studies cited in S2 Table.**
(DOCX)

**S1 Raw data.**
(ZIP)

## Acknowledgments

We acknowledge M. Carter and E. Arteca for their assistance with the literature review and graphing. The authors also thank Yuval Zelnik and two anonymous reviewers for their comments which helped substantially improve the manuscript.

## Author Contributions

**Conceptualization:** Jennifer M. Fraterrigo, Aaron B. Langille, James A. Rusak.

**Formal analysis:** Jennifer M. Fraterrigo, Aaron B. Langille.

**Investigation:** Jennifer M. Fraterrigo, Aaron B. Langille.

**Software:** Aaron B. Langille.

**Writing – original draft:** Jennifer M. Fraterrigo, Aaron B. Langille, James A. Rusak.

**Writing – review & editing:** Jennifer M. Fraterrigo, Aaron B. Langille, James A. Rusak.

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
