## [Decision Letter · Decision Letter 0]

22 Oct 2019

PONE-D-19-25590

Stochastic disturbance regimes improve prediction of ecosystem variability and dynamics

PLOS ONE

Dear Dr. Fraterrigo,

Thank you for submitting your manuscript to PLOS ONE. After careful consideration, we feel that it has merit but does not fully meet PLOS ONE’s publication criteria as it currently stands. Therefore, we invite you to submit a revised version of the manuscript that addresses the points raised during the review process.

You manuscript has been evaluated by three expert reviewers and myself. Both Reviewer #1 and #2 only raised minor points about your study, whereas Reviewer #3 has more important concerns about your study. I partially agree with Reviewer #3. One the one hand, I understand that your aim is to introduce stochasticity on some parameters that Turner's original work considers deterministic and in that sense your study provides a sufficiently novel contribution to deserve publication in PLOS One (it is not revisit of Turner's work). One the other hand, I think that some parts of the manuscript, such as the heading of the section in which you revisit Turner's model, could be modified to make this distinction as clear throughout the manuscript as it is in the Introduction. In that sense, Reviewer #3 provides very detailed suggestions. Because of this, I am inviting you to resubmit a Minor Revision of the manuscript, but I am planning to send the manuscript back to Reviewer #3 to ensure that the main four points raised in his/her report have been fully addressed.

We would appreciate receiving your revised manuscript by Dec 06 2019 11:59PM. To enhance the reproducibility of your results, we recommend that if applicable you deposit your laboratory protocols in protocols.io, where a protocol can be assigned its own identifier (DOI) such that it can be cited independently in the future. For instructions see: http://journals.plos.org/plosone/s/submission-guidelines#loc-laboratory-protocols

We look forward to receiving your revised manuscript.

Kind regards,

Ricardo Martinez-Garcia

Academic Editor

PLOS ONE

**Journal Requirements:**

**Comments to the Author**

1. Is the manuscript technically sound, and do the data support the conclusions?

Reviewer #1: Yes

Reviewer #2: Yes

Reviewer #3: Partly

2. Has the statistical analysis been performed appropriately and rigorously? 

Reviewer #1: Yes

Reviewer #2: Yes

Reviewer #3: Yes

3. Have the authors made all data underlying the findings in their manuscript fully available?

Reviewer #1: Yes

Reviewer #2: Yes

Reviewer #3: Yes

4. Is the manuscript presented in an intelligible fashion and written in standard English?

Reviewer #1: Yes

Reviewer #2: Yes

Reviewer #3: Yes

5. Review Comments to the Author

Reviewer #1: The authors present an investigation of an extension of the nonequilibrium landscape model by Turner et al. [20]. The model assumes that disturbances on the landscape have stochastic features (occurrence, size, etc...) mimicking realistic conditions. Comparing the simulation results with experimental data, the authors find a good agreement when accounting for stochasticity, in contrast with a deterministic (effective equivalent) model, which fails to predict landscape properties in many cases.

The manuscript is well written and the results are sound and well explained. However, I have minor concerns about the current version as I detail below.

1. The authors strongly emphasize that accounting for stochasticity improves prediction of the variability classification. I understand the point, but it is natural that making the model more complex would increase your predictability. This would occur for any pertinent generalization of the Turner's model. The main message is that stochastic regimes significantly impact on ecosystem variability classification. Of course, when considering it, you increase your predictability. Furthermore, the claim that 'stochasticity increases prediction' gives the idea that a stochastic resonance phenomenon is in place, i.e. stochasticity is enhancing a certain feature of the system that you are trying to measure. So, I would suggest the authors to review this terminology used in the manuscript, being careful in order to allow the results to reach with more clarity a broader audience.

2. Simple models indeed can help understand the behavior of ecosystems. But, since the proposed one assumes non-interacting units, two points come to my mind. a) I don't see how external perturbations can affect the ecosystem dynamics if the entities are non-interacting (e.g. Ising model). In my point of view, despite the fact that the variability regimes are being affected, the dynamics itself is the same (system response time properties, for instance). So, I suggest the authors to review the statement that stochastic regimes affect ecosystem dynamics (this is related to comment 1), verifying if it is really applicable in the present context. b) Analytical considerations to support the numerical results seem to be possible. In this sense, I missed basic (mean-field) relations (I would guess that the contour lines in Fig.2 are given by T = const. x S, which with y-log scale cannot be automatically understood). Besides this, it seems that a complete theory for the present model, or for other versions, could be developed, or might be already present in the literature. So, including a discussion or references about this aspect might be helpful to others understand the development on the theoretical side.

Reviewer #2: "Stochastic disturbance regimes improve prediction of ecosystem variability and dynamics" is a very interesting and well-written work.

The study focuses on an important contemporary problem and highlights the importance of considering disturbance fluctuations for a better understanding of the response of actual ecological systems.

Contents are very clear, well organized and exposed. The analysis is done carefully.

Comments:

1. Although already said that extensive analysis is left for future studies, I think it is important a longer discussion about the robustness of the results to other parameter values and degrees of stochasticity.

2. The considered dynamics probably experience transient and stationary regimes. For most of the analysis, 10000 time steps are considered, probably reaching the stationary state. Results would then be averaged over transient and stationary regimes, which have different behaviors. I think it is important to discuss this point and clarify its validity.

3. Concerning the previous point, in section (i) 100 time steps are considered, showing the transient dynamics. I propose to discuss a case with longer times.

Minor comments:

1. In line 122, "spatial extent" has not been defined previously.

2. In line 134, it is not clear how frequency is implemented.

3. In line 143, the concept of "severity" remains slightly unclear.

4. In line 176, I think it should be <=2.

5. In line 184, "width and height" is sometimes changed to "width and length". One of them should be chosen. Maybe "width and length" would be more appropriate for 2D systems.

6. Defined regions are slightly confusing. In particular, "stable" refers to both >50% and <50% of mature stage dominance. For example, In Fig. 3 regimes B and D are both described by "stable, low variance".

7. In line 276, "Comparison WITH OF published..."

8. In line 356, "spatial extent" should be "mature successional stage"?

9. In line 379, as below, the % of the expanded region can be indicated.

10. In line 417, clarify that it is the overlaying of Fig. 3?

11. In Fig. 1, stages 2,3,4,5,6 are not well distinguished with the selected dashed lines.

12. In Fig.2, yellow indicates 100% and black 0% instead of >50% and <50%, respectively.

13. In Fig.2, the deterministic case is not shown for comparison.

14. In Fig. 2, 3, S1 and S2, colorbox values are not defined in the captions.

15. In Fig. 2, 3, S1 and S2, colorbox values are difficult to read because of the small font size.

16. The analysis of Fig 4. could be done for other parameters. Maybe it can be discussed what happens for the rest of the parameters.

Reviewer #3: The manuscript by Fraterrigo et al. looks at how various forms of stochasticity of disturbances affects ecosystem dynamics using a simple theoretical methodology, and use previously published work to test how various ecosystems conform to their model. I believe that the goal of the authors is a very important one, namely better understanding of how stochasticity impacts ecosystem dynamics and predictions related to them, and trying to test this on data from various ecosystems. However, I do not think that the approach the authors took to do this is the most appropriate, and I believe it has several important flaws.

These main issues are:

1) There is some discrepancy in what do the authors aim to achieve in this paper? Is it a revisit of the classic Turner paper, or a test of the role of variability in various disturbance parameters? The way the analysis is done and presented it seems that this is a revisit of the Turner paper (e.g. Fig. 1 and its specific parameters that are the same as in the Turner paper), while the discussion and general claim of the paper is that this is an attempt to understand the role of variability. I think the authors need to choose one of these. Also, assuming that they want the second option, then a better discussion or analysis on the role of the different stochasticity sources, and in particular what happens to their combination, is quite necessary. See also points below: 13, 17 ,18, 20, 21, 25, 26, 32, 37.

2) Much of the assessment/comparison is based on the arbitrary distinction to 6 regions (A-F). Saying that something is significant because it changes the classification between these 6 regions, without explaining how they are qualitatively different or why this classification is objective, is flawed. I believe that this is an issue of the mixture between "two papers" I mention in my previous point. See also points below: 16, 24, 25, 28.

3) The comparison to data is problematic, and seems very skewed/arbitrary to authors choices of how to classify a given system. The way the authors chose to classify each system certainly needs to be better explained, but it also seems that it is quite arbitrary, and therefore why we should believe that this classification and its repercussions makes sense needs to be explained. Regardless, given this issue, and what seems like a problematic data set, it seems that the main claim of the paper, that this addition of stochasticity improves predictions, needs to be toned down. See also points below: 14, 33.

I would like to clarify some things regarding these 3 points. First, I do appreciate the choice of using a simple model, and focusing on qualitative information to test this on data. While I have some issues with the choice of model taken and its applicability, this is not in itself my main concern, but rather the discrepancy between the proposed goal of understanding stochasticity and the actual work done, which in practice mainly focuses on this models's specific results. I also appreciate that finding and collecting the data on various systems is not a simple task, and that most likely there is not enough data out there currently to preform a more detailed analysis. However, if this is the case, then this issue should be better addressed (by explicitly stating the limited power of the test preformed, toning down the strong claim (e.g. title) the authors make, and perhaps mentioning in the discussion that better data is needed).

I list below my other comments and issues, mostly organized by line numbers. While they are not major on their own, many of them make up the main issues I raised above.

4) I think in the introduction some more context is needed on previous work. In particular other works looked at similar issues of disturbance properties (e.g. Miller 2011 PNAS, Miller 2012 Eco Res), many studies looked at other system properties that are ignored here in relation to disturbances (e.g. Leibold 2004 Ecol. Lett. on dispersal, Zelnik 2019 Frontiers on local dynamics), and other models have been used to look at similar issues (e.g. Moloney & Levin 1996 Ecology, McCabe&Dietz 2019 Frontiers).

5) lines 59-71 have some strange phrasings of the topic - line 59: "the response of ecosystem variability", line 61: "disturbance-driven changes in the variability of a response can indicate differences in", line 71: " altered ecosystem variability". I guess you mean to talk about about "changes in variability", how they might matter or inform us, etc. But this is currently unclear.

6) lines 73-75: It is not true that there are 3 relevant properties of disturbances (frequency, severity & extent). For instance, species composition (i.e. how different species are sensitive to a given disturbance) might matter significantly, see for instance Arnoldi et al. 2018 JTB.

7) lines 83-89 - The purpose of this text is not clear, and seems like it is mostly unnecessary.

8) line 152 - The authors themselves defined disturbance frequency in a previous publication "It is widely recognized that disturbance frequency, defined as the mean number of events per time period...", in a way that is not consistent with talking about "stochastic frequency" versus "constant frequency", since both would be called frequency in any case. It would be better to talk about return interval for this definition, if not elsewhere in the manuscript.

9) lines 153-162 this seems ill defined. Are you really always having 16 events? In this case, you should better describe your randomization process (is this a point process?)? Or do you just have 16 disturbances on average? (which would make more sense)

10) lines 196-200 seem counter productive - there is stochasticity in all these processes (even those controlled by humans), it is more the question of how much variation exists, and therefore, how much it might matter that they are stochastic. Trying to untangle the roles of these different forms of stochasticity, and their combination, is certainly a good idea, but the logic of doing so should be stated clearly.

11) lines 207-208 and lines 249-250 give very different runtime numbers for simulations - why?

12) line 286 - what does it mean that "We focus on variance"? If it is simply that the mean doesn't change much between your 4 models, why not say that explicitly?

13) lines 292-304 - is there anything different between what you did for the model runs here and for the T-S parameter space? I guess not, and if so, this whole explanation is cumbersome and unnecessary, since as I understand it, you basically estimated where the different systems (from publications) are on the T-S parameter space, given values of S and T for each system.

14) line 311-313 - How is this classification done? This is a critical point, to which you basically give no information, which is particularly noteworthy considering you give plenty of (often too much) information about the specific model choices and technicalities.

15) lines 319-320 are strange. You say you measure S and T to "facilitate interpretation"? But, without estimating S and T, no comparison can be done - is that not so?

16) lines 331-334, at least on their own, seem unsubstantiated. Why are differences in variance levels indicating "distinct ecosystem dynamics". No qualitative shift, if one exists, is mentioned. Also, why were this specific parameter set chosen, besides the fact that it was used in the original Turner paper? How general is the result that averages do not change much, but variances do?

17) lines 336-341 - Overall I would say that Fig.1 chooses parameter that are not very representative. Given this specific frequency choice, disturbances cannot interact, but this is not an obvious choice. For any frequency that is higher interactions are likely to occur, and this should be mentioned.

18) lines 346 and 352, "indicating high ecosystem stability" and "indicating reduced ecosystem stability" - what is meant by stability here? If it is the lack of variability, then this is correct, except there's no need for "indicating" - it is a fact. If it is something else, then that needs to be better explained.

19) lines 347-364 are mostly stating things that are quite evident. I think they can be shortened or clarified.

20) lines 368-369 - but how much is the variance compared to the "extra" variance from all 3 separately?

21) lines 371-377 are again almost trivial. A multiplication of T*S would tell me the baseline values I would expect for the mature stage. What would be interesting to know is how it drifted from this baseline.

22) lines 378-379 a bit vague, and not something that is apparent from the figure itself. How much actually changed. And can it be accounted for by "adding" the 3 separate components?

23) lines 386-389 appear to be wrong. Did you mean that the areas below and above 50% are separated by the white line?

24) lines 391-396 Amplified by how much? Perhaps a map of this would be more illuminating then some of these other measures that use arbitrary classifications.

25) lines 397-399 comparing the fully stochastic scenario, with extent of frequency scenarios, region C grows less for the fully stochastic, but region E grows more. Is this because they are related to each other, or is something else happening?

26) lines 402-407 tell us things we could see from before, which is fine. But what about an explanation of what is happening? In particular, is the severity low variance related to the specific choice where it can only be lower for the stochastic scenario? What if you compared a more appropriate scenario?

27) line 413 - "major transitions in ecosystem dynamics", again the claim that this represents major transitions seems unsubstantiated.

28) lines 418-420 - It seems that you are saying that near the "boundaries" between your regions (A-F), there are more changes between regions due to addition of different sources of stochasticity. This is to a large extent a form of tautology, since changes near an edge of a region are more likely to lead to a different region due to geometric considerations alone. Or did you mean something else?

29) lines 427-429 - this should be given in context. It is an artifact of the parameter space, e.g. stochasticity means less when disturbances occur very infrequently (no interactions between disturbances) or when they are very frequent (system is almost always at the lowest stage).

30) lines 439-441 - how is this different than before? Throughout the paper you test 4 different stochastic models, and it seems that here as well. So what is different about this? Also, it is not clear what parameter you used here to get Fig. 5.

31) lines 442-445, I guess you meant "f" instead of "T"?

32) lines 439-451 - key insights are missing here. The fact that for T>8 (small f) you get a larger difference between the deterministic and stochastic models is not very surprising, considering that for the deterministic model you cannot get interactions of disturbances, but construction. when T<8 (high f) this changes, and things become more comparable. Interestingly, this is exactly when the extent-only model has larger variance than others, which is consistent with the fact that it doesn't have interactions for T>8 (i.e., the extent-only model "should" get larger values overall, but for T>8 this is not seen). Also, why is the peak around T=1? Is that a general result?

33) lines 467-476 - over 60% of papers were with disturbance size=1. This seems quite problematic for the analysis, and at the very least should be mentioned.

34) lines 502-505 - are you trying to say that stochasticity matters, and that variability and stochasticity are related? That alone seems redundant. If you are saying something else, it should be better clarified.

35) lines 506-519 seem to just repeat results and conclusions from the original Turner paper. And lines 519-527 are quite strange and seem out of place. In particular, talking about dispersal while it is not mentioned at all in the whole paper, would only make sense if this was to discuss how adding dispersal would affect the results - and this is not what the authors do here.

36) lines 543-550 seem to try and make several points, which seem either problematic or that I could not follow properly. In the first sentence you say "in accordance" - of what to what? Are you just saying that the addition of variance due to stochasticity in frequency and extent lead to larger regions ("stable, high or very high variance")? Why does this "suggest" that there is increased fluctuations? And regardless, why not check that explicitly? Also, why did "ecosystem dynamics remained stable..."? Stable in what sense? And how does this indicate that stochasticity does not need to drive systems to alternative states? Certainly, that statement as is (I would rephrase it as "stochasticity will not always drive systems to collapse in finite time") is true, but it is also widely known, and rather, any divergence from it would be surprising. Finally, one point that I suppose you tried to make, but I am not at all sure, is that

37) lines 562-572 point out to a major modeling problem, which is that the comparison of deterministic vs. stochastic severity of disturbances was not done properly. The last sentence calls on more research in this direction, but I think that the minimal thing you needed to do was test this directly, even if in a limited fashion (e.g. compare deterministic model with severity 8 to a stochastic one where severity is randomly chosen between 15 an 1)

38) lines 586-606 - I think it is here that you need to mention the effect of dispersal, that your model does not account for, similarly to how you do this with issues of internal dynamics.

39) Figure S1 does not use colors, and the color axis is not the same for all panels, making visual comparisons difficult.

40) "Turner et al. [20]" is used many many times throughout the manuscript. This can often be shortened, e.g. "deterministic model"

41) line 71 - "Disturbance" should be "A disturbance" or "Disturbances", no?

42) line 253 - typo: "produced by using"

43) line 276 - typo: "Comparison with of published"

6. PLOS authors have the option to publish the peer review history of their article (what does this mean?). If published, this will include your full peer review and any attached files.

Reviewer #1: No

Reviewer #2: No

Reviewer #3: Yes: Yuval Zelnik

---

## [Author Response · Author response to Decision Letter 0]

22 Jan 2020

See attached file "Response to Reviewers" for responses to specific comments.

---

## [Decision Letter · Decision Letter 1]

19 Feb 2020

Stochastic disturbance regimes alter patterns of ecosystem variability and recovery

PONE-D-19-25590R1

Dear Dr. Fraterrigo,

We are pleased to inform you that your manuscript has been judged scientifically suitable for publication and will be formally accepted for publication once it complies with all outstanding technical requirements.

With kind regards,

Ricardo Martinez-Garcia

Academic Editor

PLOS ONE

Additional Editor Comments (optional):

Reviewers' comments:

Reviewer's Responses to Questions

**Comments to the Author**

1. If the authors have adequately addressed your comments raised in a previous round of review and you feel that this manuscript is now acceptable for publication, you may indicate that here to bypass the “Comments to the Author” section, enter your conflict of interest statement in the “Confidential to Editor” section, and submit your "Accept" recommendation.

Reviewer #3: All comments have been addressed

2. Is the manuscript technically sound, and do the data support the conclusions?

Reviewer #3: Yes

3. Has the statistical analysis been performed appropriately and rigorously? 

Reviewer #3: N/A

4. Have the authors made all data underlying the findings in their manuscript fully available?

Reviewer #3: Yes

5. Is the manuscript presented in an intelligible fashion and written in standard English?

Reviewer #3: Yes

6. Review Comments to the Author

Reviewer #3: I have read the revised manuscript by Fraterrigo et al. and I find it to be a significant improvement over the original submission. The paper reads more easily, the introduction and discussion are more appropriate to the results, and overall the paper's contribution is more clear. I am therefore happy to recommend it for publication.

I have a few minor suggestions, which the authors may find useful, that I write below.

Line 373, "Stochastic severity generally caused more modest changes in variance compared to other stochastic attributes (Fig 3).", I think you should say that this is result of the decrease of average severity (as you basically later show in lines 465-470)

Comparing Fig.2 and Fig.3, I cannot visually see that there are big changes in variance patterns (Fig.3), and not so much in average cover (Fig.2). I don't think this is necessarily a major point, but I would recommend either clarifying (visually or otherwise) why/how the variance changes are more substantial than those for average cover, or putting less emphasis on this point.

If possible, it would be great if you could clarify further how you classify into the 4 categories. Looking more into this, I realize that in practice you see 3 categories in the studies you found. I think it would be very useful if you therefore clarify the "practical procedure" for each such category (e.g. "If we see that there has been a major change in cover and also X, than we note this as category Y").

In the legend of Fig.6, you write "(e)" instead of "(d)"

A minor note, you spelled my name in the Acknowledgments as Zelnick instead of Zelnik.

7. PLOS authors have the option to publish the peer review history of their article (what does this mean?). If published, this will include your full peer review and any attached files.

Reviewer #3: Yes: Yuval Zelnik

---

## [Editor Report · Acceptance letter]

24 Feb 2020

PONE-D-19-25590R1 

Stochastic disturbance regimes alter patterns of ecosystem variability and recovery 

Dear Dr. Fraterrigo:

I am pleased to inform you that your manuscript has been deemed suitable for publication in PLOS ONE. Congratulations! Your manuscript is now with our production department. 

With kind regards,

on behalf of

Dr. Ricardo Martinez-Garcia 

Academic Editor

PLOS ONE